# Structural Characterization of Several Cement-Based Materials Containing Chemical Additives with Potential Application in Additive Manufacturing

**DOI:** 10.3390/ijms24097688

**Published:** 2023-04-22

**Authors:** Alexandru Florin Simedru, Anca Becze, Oana Cadar, Daniela Alexandra Scurtu, Dorina Simedru, Ioan Ardelean

**Affiliations:** 1Department of Physics and Chemistry, Technical University of Cluj-Napoca, 400114 Cluj-Napoca, Romania; 2INCOD-INOE2000, Subsidiary Research Institute for Analytical Instrumentation Cluj-Napoca, 400293 Cluj-Napoca, Romania

**Keywords:** additive manufacturing, cement accelerators, cement-based material, chemical additive, hydration, XRD, SEM-EDX, LF-NMR

## Abstract

The rapid increase in additive manufacturing applications in all industries has highlighted the lack of innovative technologies and processes in the construction industry. Several European and international policies are in place to guide the development of the technological processes involved in the construction industry toward a sustainable future. Considering the global concerns regarding this industry, the purpose of this study was to develop new cement-based materials that are capable of accelerated hydration and early strength development for use in additive manufacturing. Ca(NO_3_)_2_·4H_2_O, Al_2_(SO_4_)_3_·18H_2_O and Na_2_S_2_O_3_·5H_2_O were used to obtain the accelerating effect in the hydration of Portland cement. Based on results obtained from X-ray diffraction (XRD), scanning electron microscopy and energy-dispersive X-ray spectroscopy (SEM/EDX) techniques, as well as low-field nuclear magnetic resonance relaxometry (LF-NMR) techniques, it was demonstrated that all accelerators used have a quickening effect on cement hydration. The addition of Na_2_S_2_O_3_·5H_2_O or combined Na_2_S_2_O_3_·5H_2_O and Ca(NO_3_)_2_·4H_2_O led to obtaining new cement-based materials with early strength development and fast hydration of microorganized internal structures, critical characteristics for 3D printing.

## 1. Introduction

The construction industry makes an essential contribution to the economic growth of a country, but it also has a significant impact on the environment due to the generated pollution. Although the COVID-19 pandemic had a negative impact on the development of the construction industry, in 2022, it generated 8.2 trillion USD, with an estimated compound annual growth rate (CAGR) of 7.3% [1] and 7% to 8.5% of global employment [2]. Regarding the environmental issues related to the construction industry, it is considered to have the most serious environmental impact among all industries on air, water, soil, and noise pollution, waste generation, and greenhouse gas emissions [3]. The World Business Council for Sustainable Development (WBCSD) and its partners proposed several guidelines for the sustainable transformation of construction industry at the Stockholm+50 international meeting in June 2022 [4]. Among these guidelines, one focuses on material efficiency and innovation [5]. The recent trends in construction materials follow this guideline to maximize process efficiency and reduce cost by: introducing polymer-derived fibers into concrete to make it bendable; using softwood to reduce emissions and waste in the manufacturing process; continuing to use recycled materials; and using 3D printing to reduce material losses, reduce manpower and execution time [6].

Additive manufacturing (AM) or 3D printing derives from stereolithography (SLA) is a process developed in 1970, but introduced by Chuck Hull in 1984 [7]. SLA creates 3D objects from liquid photopolymer resin, layer by layer, using a light-emitting device [8], and 3D printing uses a computer-controlled printhead to build different shapes using a particular volume of a specific material in sequential layers following a 3D computer-aided design (CAD) model [9,10,11]. Since the beginning of AM, many industries, such as aerospace, automotive, and medical, have developed and implemented important applications [2,12] for sustainable transformation of these sectors, but the construction industry still lags, although a study identified that in the period 1997–2016, there were 4117 publications in Web of Science and 7173 in ScienceDirect directly related to 3D printing in building and construction [9]. The immediately recognized benefit of using AM in the construction industry will be the elimination of formwork, which translates directly into a reduction in cost, materials, labor, execution time, produced waste, and increased freedom design [13]. Several challenges (social, economic, and technical) must be overcome in order to unlock the previously mentioned benefits. Some of them are related to the conservativism from the construction industry and the low investments in technology [2,13], and others to the materials involved [14,15]. Today, six main AM technologies used in the construction industry—selective binding, extrusion, contour crafting, binder jetting, shotcrete, and slipform-based technologies [12,15]—address these challenges.

The main issues associated with these technologies relate to the cement material used in layer-by-layer AM processes. The layers of material should be hard enough to support subsequent layers, set quickly enough to allow reasonable working time, and adequately bond previous and subsequent layers. One way to achieve these goals is to modify cement, one of the basic components of concrete, to increase early strength and reduce setting time by introducing an accelerator [16]. As accelerators of hydration of Portland cement, cations and anions have the following orders of effectiveness: Cl^−^ > S_2_O_3_^2−^ > SO_4_^2−^ > NO_3_^−^, Cl^−^ > Br^−^ > I^−^ and Ca^2+^ > Mg^2+^ > Na^+^ [17]. Among them, calcium nitrate Ca(NO_3_)_2_, aluminum sulfate Al_2_(SO_4_)_3,_ and sodium thiosulfate pentahydrate Na_2_S_2_O_3_·5H_2_O are used as accelerators in concrete and concrete applications, but some of them are less studied [18,19,20]. Other studies report the obtaining of AM materials by combining ordinary Portland cement with calcium aluminate cement or calcium sulfoaluminate cement [21,22]. Shakor et al. uses lithium carbonate (Li2CO3) as an accelerator in one study, whereas Prasittisopin et al. uses calcium aluminate cement or calcium sulfoaluminate cement in another. Their tests demonstrate the increase of compressive strength, the improvement in thermal properties, and the dependence of these properties on the shape and size of the tested material [21,22].

Generally speaking, cement paste is composed of four phases: cement paste, surface products (products deposit on unreacted cement particles such as C-S-H), pore products (polycrystalline phases form in the water-filled pore space between cement particles such as calcium hydroxide), and capillary pores [23]. To follow the phases and the structural network changes in Portland cement caused by the addition of accelerators, several techniques can be used, such as X-ray diffraction (XRD), scanning electron microscopy with energy-dispersive X-ray spectroscopy (SEM/EDX), and low-field nuclear magnetic resonance relaxometry (LF-NMR) [24,25,26,27,28,29,30,31,32,33,34,35,36]. XRD is a nondestructive material characterization technique for studying the fine structure of matter, and is successfully used as an alternative to other techniques for studying hydration compounds and quantitative crystalline phases [24,25,26,27,28]. SEM/EDX is a nondestructive technique (when used without coating) used to study and analyze a material’s morphology, topography, chemical composition, particle orientation, and crystallographic information, allowing us to obtain valuable information about the structure of cement-based materials, such as the hydration of the cement, formation and distribution of hydration products, and the homogeneity of the cement paste [29,30,31,32,33]. LF-NMR is a noninvasive and nondestructive quantitative technique that provides valuable information on hydration kinetics, porosity, and pore size distribution [27], and has been used in cement and concrete research for many years [34,35].

Much work is still needed to find fast, efficient, cost-effective, and environmentally friendly solutions to overcome the shortcomings of AM in the construction industry. This work is being conducted in response to the need to find suitable cement-based material that can meet the needs of the emerging construction industry. Five new cement-based materials using Ca(NO_3_)_2_·4H_2_O, Al_2_(SO_4_)_3_·18H_2_O and Na_2_S_2_O_3_·5H_2_O and different combinations thereof were prepared, tested, and compared with white cement 52.5R by XRD diffraction, SEM/EDX, and LF-NMR to provide information about their structure and their role as accelerators in cement-based materials.

## 2. Results and Discussion

The samples were analyzed by X-ray diffraction. Figure 1 shows the X-ray diffraction patterns and their assignments using the abbreviations listed in Table 1. The diffraction patterns of all investigated samples show narrow peaks characteristic of several crystalline mineral phases identified by their PDF (powder diffraction file).

Table 2 and Figure 2 show the results of the semiquantitative analysis of the mineral phases present in the samples and their degree of crystallinity (the ratio between the integrated area of crystalline species and the integrated area under the overall XRD profile or sum of crystalline and amorphous species in arbitrary units).

The results of the X-ray diffraction show different variations in C_3_S, C_2_S, CC and CH content when accelerators are introduced to Portland cement. Some of the interpretations are presented below.
The accelerators inhibit C_3_S hydration when added to Portland cement. The addition of Na_2_S_2_O_3_ or Ca(NO_3_)_2_ simultaneously with Na_2_S_2_O_3_ and Al_2_(SO_4_)_3_ results in an increase in C_3_S content over Ca(NO_3_)_2_. The highest value for C_3_S content is obtained for CPNa, while the lowest value is assigned to CPCa. It is well known that Na compounds are very reactive, especially when combined with water [37], accelerating hydration and increasing the amount of hydration products [23]. Rapid hydration of the cement can lead to early freezing of the cement and thus an increase in the C_3_S content.The addition of accelerators to Portland cement contributes to a reduction in the content of C_2_S supporting its hydration. The highest value is obtained in the case of CP2CaNa and decreases for the samples CPCa, CCPaNa, and CPNa, which will have similar values, while the lowest value is obtained in the case of CCPaNaAl.Regarding the reaction products C-H and CC, the following behavior can be observed in the samples containing the accelerator. The content of C-H increases for CPCa, with the lowest value being obtained for CPCaNaAl where the presence of ettringite (AFt) has been detected. The appearance of Aft in Portland cement with the addition of Al_2_(SO_4_)_3_ is due to the increase in sulfate content in the liquid hydrate phase after mixing, which promotes the production of acicular ettringite crystals [38].In the case of CC, all the values obtained are below the value obtained for CP. The highest value is obtained in the case of CPNa and the lowest in the case of CPCa. Based on these results and the crystallinity degree, it can be concluded that the addition of Ca(NO_3_)_2_ in Portland cement increases crystalline C-H, increasing the crystalline degree, while the addition of Na_2_S_2_O_3_ has the opposite effect. Calcium silicate hydrate (C-S-H) is the main hydration product of Portland cement, formed by the reaction of C_3_S and C_2_S with water. It is generally amorphous or weakly crystalline [26,39]. A decrease in C_2_S and a sharp decrease in the degree of crystallinity in the case of CPNa may indicate that the addition of Na_2_S_2_O_3_ in CP leads to a larger amount of C-S-H hydration as a result of C_2_S hydration than in the rest of the analyzed samples.The C_3_S/C_2_S ratio, which expresses the rate of cement hardening, is shown in Figure 3. It can be seen that the addition of Na_2_S_2_O_3_ accelerates the cement hardening process the most and the addition of Ca(NO_3_)_2_ the least. The addition of Al_2_(SO_4_)_3_ accelerates the cement hardening process more than Ca(NO_3_)_2_ when comparing the two samples containing the same content of Na_2_S_2_O_3_, CP2CaNa and, respectively, CPCaNaAl.


The samples were examined by scanning electron microscopy (SEM) with energy-dispersive X-ray spectroscopy (EDX). The SEM analysis provided information about the surface topography and morphology. The study of the recorded images (Figure 4) shows that: CP and CP2CaNa have porous surfaces with randomly distributed conglomerates; CPCa and CPNa show compact surfaces with randomly distributed small formations and cracks, indicating rapid hardening of the samples; and CPCaNa and CPCaNaAl show relatively compact, slightly porous surfaces and a high number of pores randomly distributed on the surface.

The distance between the pores and their dimensions were measured and are shown in Table 3. The minimum value for the pore diameter was for CPCa sample (~2.88 μm) and the maximum for CP (~9.44 μm). These values classified the pores observed in the investigated samples as air holes (>several μm), although they are very close to large capillary pore values (100–1000 nm) [40]. The addition of accelerators in CP leads to a reduction in pore diameter, indicating a more compact structure in these samples. The above idea is also supported by the values obtained for the pore spacing. The addition of accelerators in CP leads to a reduction in pore spacing: a small pore spacing indicates a high pore density [41].

The distribution of the elements on the sample surface and the change in chemical composition on the sample surface can be observed by analyzing the element map.

Figure 5 shows the surface and the distribution of elements on the analyzed surfaces expressed in “false” colors. Table 4 shows the elemental composition obtained from the EDX analysis.

According to the results, O and Ca have the highest concentration in all samples. The Na content increases with the increase of Na_2_S_2_O_3_ content, albeit sometimes unevenly. A direct connection between Ca and Si can be observed. As the Ca content increases, the Si content decreases. Although one might expect that the addition of Ca(NO_3_)_2_ to Portland cement would increase the Ca content at the surface of the samples, the results suggest the opposite, indicating that increasing the Ca content leads to structural modification in the interior and at the surface of the samples. The ratios of Ca/Si and Ca/(Si+Al) exceed the values reported in the literature for calcium silicate hydrates [42], indicating the existence of Ca-containing phases alongside C-S-H. These results are consistent with the changes in the mineral phases of CH, SS, C_2_S, and C_3_S observed in the XRD results, suggesting that the addition of Ca(NO_3_)_2_ in Portland cement increases the formation and the content of Ca-dominant phases, like Ca(OH)_2_ and/or CaCO_3_ in the examined samples.

The NMR analysis consists of monitoring the hydration responses of the samples over 6 hours by recording the echo series every 15 min throughout the hydration period (Figure 6). As the initial slope of echo intensity increases with echo time, it becomes obvious that the addition of accelerators leads to rapid hydration of the cement mass, particularly the addition of combined Ca(NO_3_)_2_·4H_2_O and Na_2_S_2_O_3_·5H_2_O.

The distribution of transverse relaxation times versus hydration time was plotted (Figure 7). The relaxation time distributions were extracted using a numerical Laplace transform from CPMG echo series.

According to Figure 7, two peaks were observed and were assigned to the presence of two aqueous environments. The first peak, which occurs at the beginning of the hydration phase, is due to the chemically bounded water and occurs when a certain amount of water (flocculant water) combines with the cement grains and forms multiple microorganized systems (cement grain flocculation, chemical reactions, ettringite pores). The second peak is due to the water (capillary water) filling the space between these microstructures [43] or from the surface of the sample [44]. For the samples examined, the first peak occurs after 1 ms and moves slightly to lower values (below 1 ms) during the hydration process. The second peak appears after 10 ms and moves slightly to lower values (below 10 ms) during the hydration process. This result indicates that the pores in the samples are becoming smaller due to the hydration reactions. A part of the capillary water participates to the formation of microorganized structure or evaporates.

The plot of peak area versus hydration time shows an initial decrease in area of the first peak (Figure 8) for all samples examined. CP begins a slow rise from minute 195, while samples containing accelerator show an accelerated increase earlier. The second peak continuously decreases for all samples, indicating a decrease in the amount of free water in the capillary pores. Slower decay can be observed for CP than for accelerator-containing samples. These results imply that the hydration process is relatively constant at the beginning of the water-cement interaction, followed by an increase in flocculant water and a decrease in capillary water. This behavior may be due to the participation of capillary water in the process of forming new cement-based structures.

In Figure 9, the maximum point of the T2 relaxation time is plotted for each of the analyzed samples, showing a slow decrease for the CP and CPCaNaAl samples and an accelerated decrease for all other samples. The T2 evolution of CP and CPCaNaAl indicates a slow decrease of capillary water, indicative of a slow change in pore volume, whereas for the other samples analyzed, capillary water decreases rapidly, indicative of a rapid and significant change in pore volume.

Based on the data obtained by NMR analysis, it can be stated that the hydration process of the analyzed samples, or rather the change in pore volume and the formation of organized structures, occurs fastest in CPNa samples, followed by CPCa, CPCaNa, CP2CaNa, CPCaNaAl and CP.

## 3. Materials and Methods

### 3.1. Sample Preparation

White cement 52.5R produced in Romania (composition: 95–100% clinker and 0–5% other components) and accelerators (Ca(NO_3_)_2_·4H_2_O, Al_2_(SO_4_)_3_·18H_2_O and Na_2_S_2_O_3_·5H_2_O) from Merck (Darmstadt, Germany) were used for the preparation of the experimental cement-based materials. The samples were prepared with the same water/cement ratio of 0.4 and different proportions of accelerator (1–3%) according to Table 5.

The level of the accelerator to be used for this study was determined by multiple tests ranging from 1% to 5% accelerators. In order to preserve the cement paste’s properties, the accelerator concentration range was set between 1% and 5%. Further study was conducted on concentrations that formed compact materials at the smallest concentrations.

The chemical analyses were conducted on the obtained materials at an age of 90 days, choosing this age to ensure that the hydration process was complete.

### 3.2. Experimental

#### XRD Analysis

X-ray diffraction was performed using a Bruker D8 Advance powder diffractometer (Bruker, Karlsruhe, Germany) with a LynxEye XE detector and Cukα1 radiation, λ = 1.54060 Å at room temperature. The following measurement conditions were applied: operating tension 40 kV; current 40 mA; scanning speed 0.02°/s; scanning range 2θ = 10–80°. The crystalline phases were identified using Bruker DIFFRAC.SUITE 7.5 software, and crystallinity, and semiquantitative percentages of identified phases were estimated using Bruker DIFFRAC.SUITE EVA 3.0 software.

### 3.3. SEM-EDX Analysis

The sample surface (surface topography, pore size and elemental composition) was examined using scanning electron microscopy (VEGA3 SBU, Tescan, Brno-Kohoutovice, Czech Republic) with energy dispersive X-ray spectroscopy (Quantax EDS, Bruker, Karlsruhe, Germany) SEM/EDX at room temperature. The analysis of the EDX data was carried out with Esprit 2.2 software. Small specimens of ~2 mm^2^ were used for this study. The magnification level was between 350–420 x. No special treatments such as gold coating were applied to these samples.

### 3.4. Low-Field NMR Relaxometry Analysis

Transverse relaxation measurements were performed with a Bruker MINISPEC MQ20 NMR Analyzer, using the well-known Carr–Purcell–Meiboom–Gill (CPMG) technique [35,36]. The evolution of water consumption in the capillary pores was monitored by an experiment carried out under the following conditions: 1000 echoes were recorded in the CPMG pulse sequence, with an echo time of 0.08 ms, and 32 scans/experiment.

## 4. Conclusions

The aim of the study was to obtain and characterize several cement-based materials with white cement and accelerators—Ca(NO_3_)_2_·4H_2_O, Al_2_(SO_4_)_3_·18H_2_O and Na_2_S_2_O_3_·5H_2_O—in different concentrations, and to test their potential for usage in 3D printing. The obtained materials were studied using the following techniques: X-ray diffraction, scanning electron microscopy with energy-dispersive X-ray spectroscopy (SEM/EDX), and low-field nuclear magnetic resonance relaxometry (LF-NMR).

The X-ray diffraction (XRD) patterns of the obtained materials present crystalline mineral phases characteristic of cement materials (CH, CC, C2S, C3S and Aft). Their content modifies with the type of accelerator used. C3S/C2S ratio was used to observe the hardening speed of the samples and significant increases in the materials containing accelerators compared to Portland cement.

Scanning electron microscopy coupled with energy-dispersive X-ray spectroscopy (SEM-EDX) revealed information regarding the surface of the analyzed materials. Significant differences were observed when comparing Portland cement with samples containing accelerators in pore size, the distance between pores, and the spatial distribution of the elements on the surface of the material.

Low-field nuclear magnetic resonance (LF-NMR) relaxometry was used to observe the water kinetics in the obtained materials over 6 h. The results show a decrease in the volume of pores for materials containing accelerators compared to Portland cement and faster hydration of these materials.

The obtained results confirm the potential of the obtained materials for use in 3D printing and suggest that cement paste with Na_2_S_2_O_3_·5H_2_O as accelerator is the most suitable of them for the intended purpose.

## Figures and Tables

**Figure 1 ijms-24-07688-f001:**
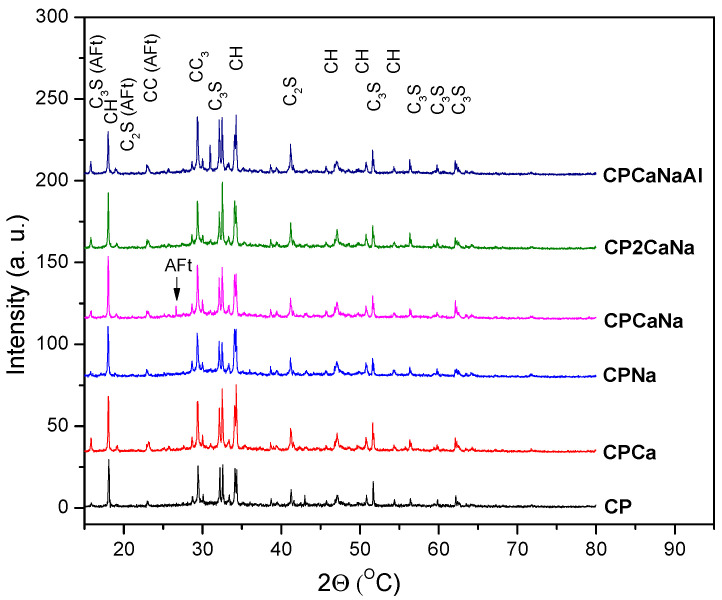
X-ray diffraction patterns and diffraction peak assignments.

**Figure 2 ijms-24-07688-f002:**
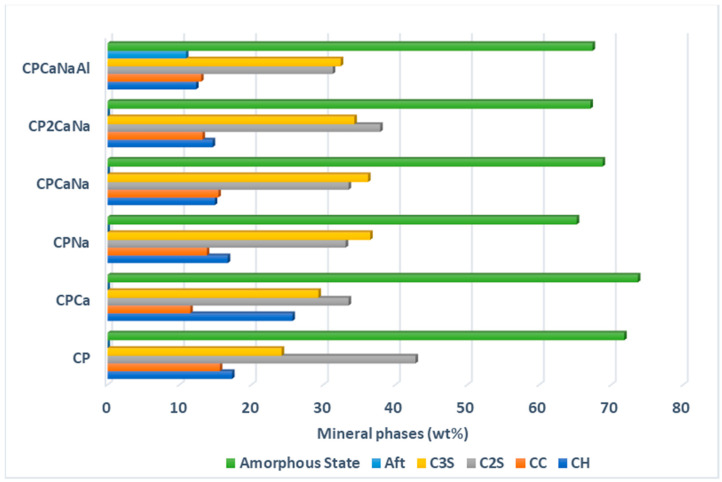
Graphical representation of mineral phase content and degree of crystallinity in tested samples.

**Figure 3 ijms-24-07688-f003:**
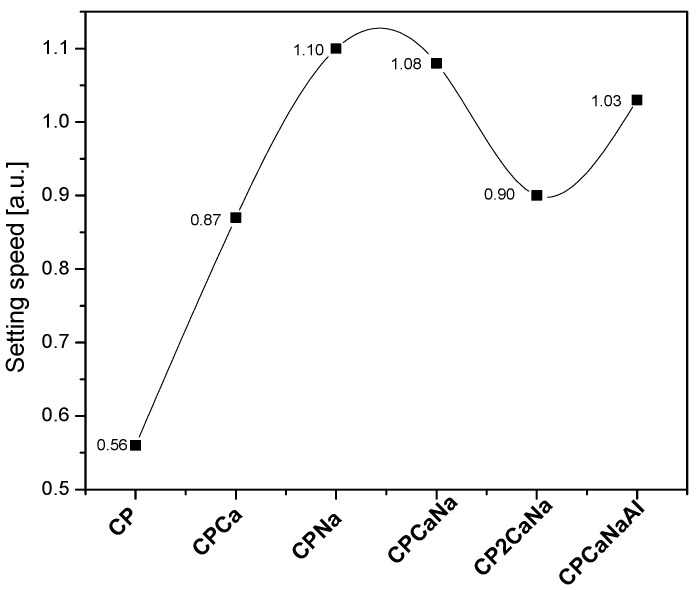
Setting speed of tested samples expressed as a C3S/C2S ratio.

**Figure 4 ijms-24-07688-f004:**
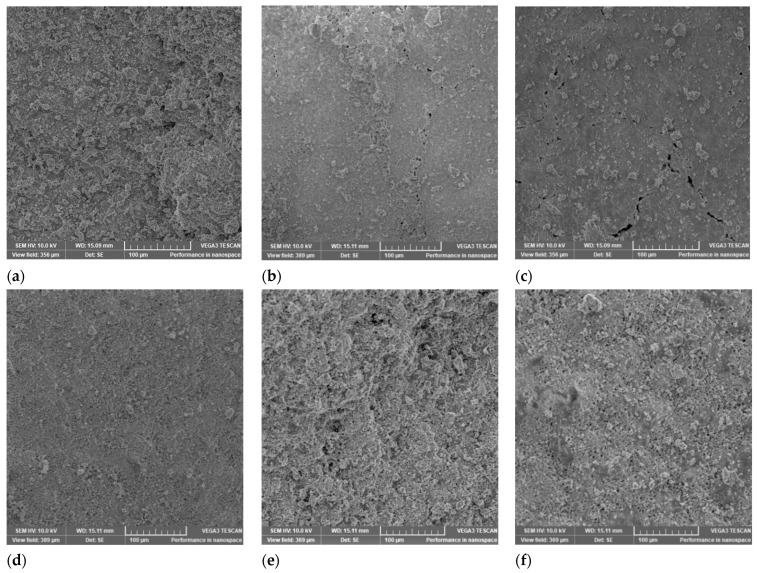
Surface of: (**a**) CP; (**b**) CCPa; (**c**) CPNa; (**d**) CPCaNa; (**e**) CP2CaNa; (**f**) CPCaNaAl by SEM analysis.

**Figure 5 ijms-24-07688-f005:**
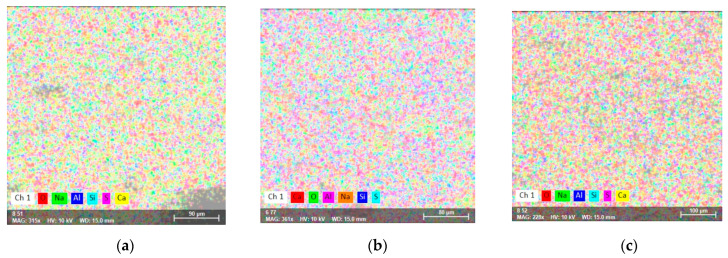
SEM-EDX map surface of: (**a**) CP; (**b**) CPCa; (**c**) CPNa; (**d**) CPCaNa; (**e**) CP2CaNa; (**f**) CPCaNaAl.

**Figure 6 ijms-24-07688-f006:**
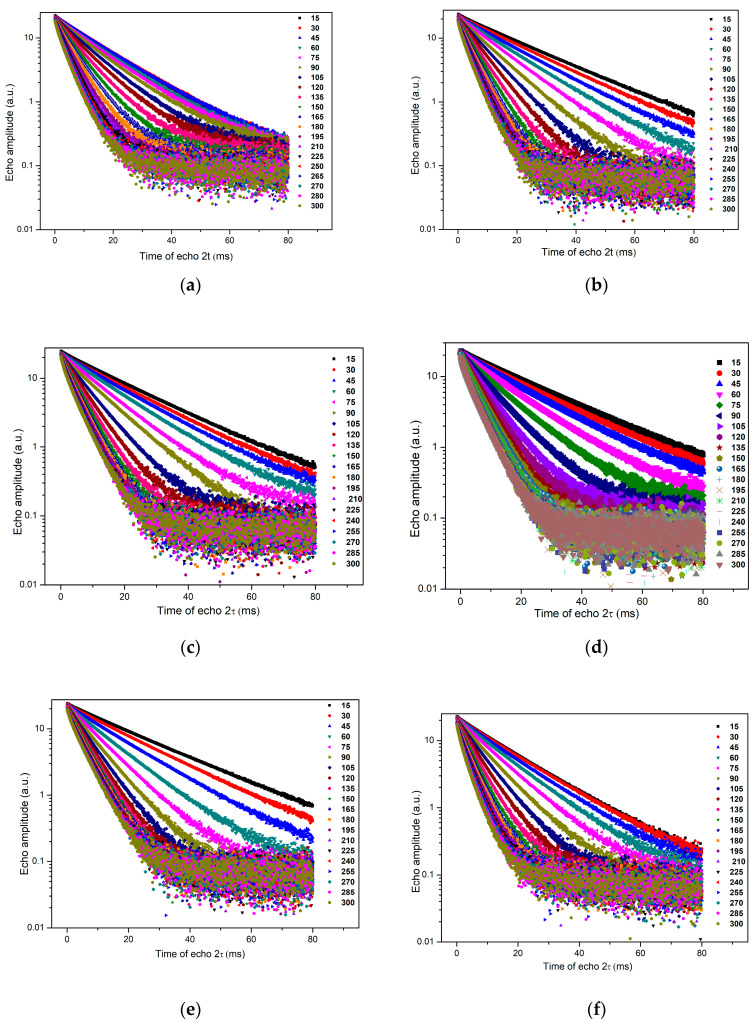
CPMG echo series for: (**a**) CP, (**b**) CPCa, (**c**) CPNa, (**d**) CPCaNa, (**e**) CP2CaNa and (**f**) CPCaNaAl.

**Figure 7 ijms-24-07688-f007:**
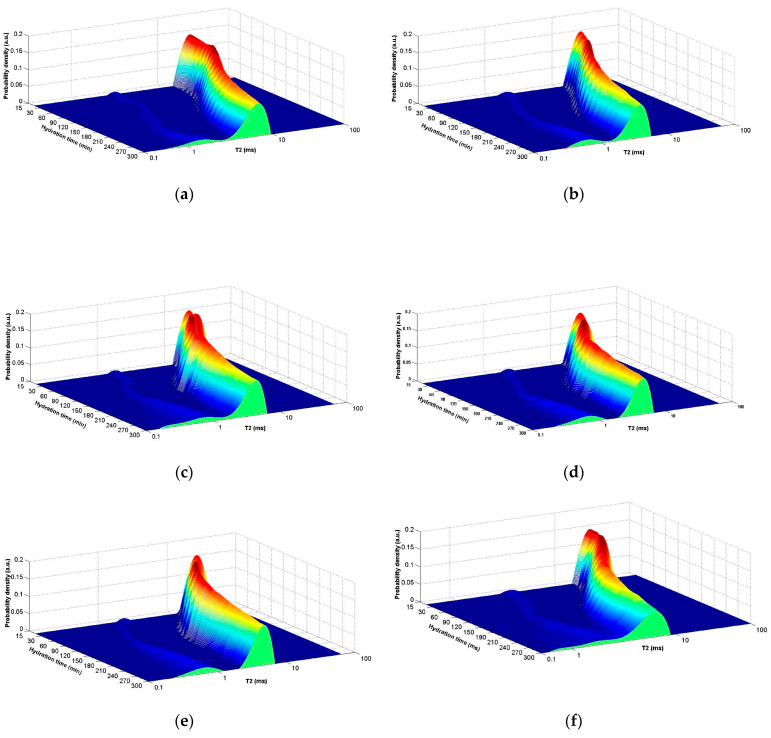
Distribution of relaxation time T_2_ as a function of hydration time for (**a**) CP, (**b**) CPCa, (**c**) CPNa, (**d**) CPCaNa, (**e**) CP2CaNa and (**f**) CPCaNaAl.

**Figure 8 ijms-24-07688-f008:**
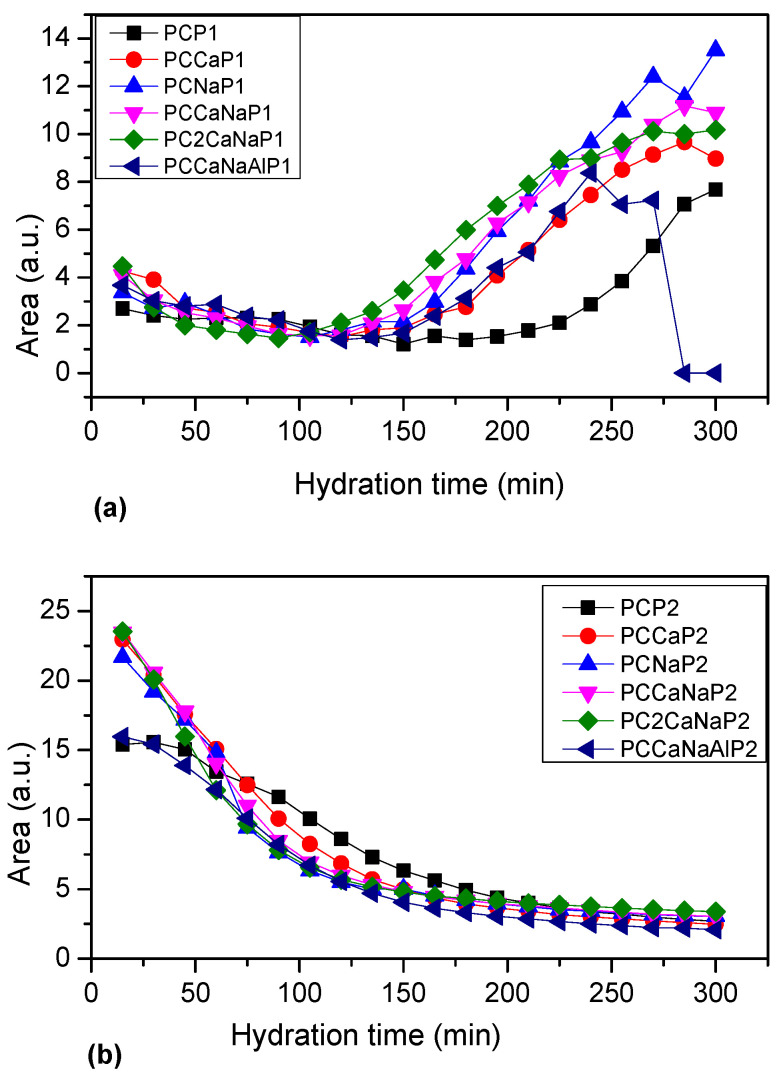
The plots of area versus hydration time for the two peaks characteristic to the aqueous environments in cement. (**a**) P1-first peak, (**b**) P2-second peak.

**Figure 9 ijms-24-07688-f009:**
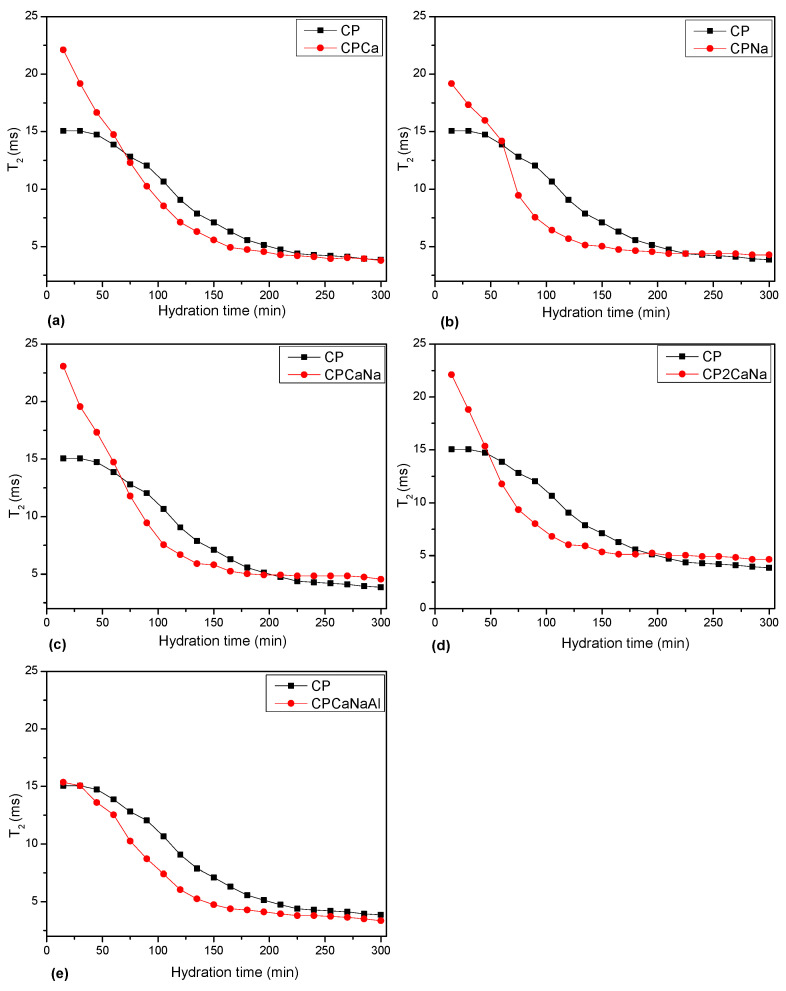
T2 maximum value versus hydration time for the investigated samples: (**a**) CPCa, (**b**) CPNa, (**c**) CPCaNa, (**d**) CP2CaNa and (**e**) CPCaNaAl compared with CP as indicated in the legend.

**Table 1 ijms-24-07688-t001:** The mineral phase identified and the parameters used for their identification.

Mineral Phase	Chemical Formula	Abbreviation	PDF
**Portlandite**	Ca(OH)_2_	CH	PDF 00-004-0733
**Calcite**	CaCO_3_	CC	PDF 01-083-1762
**Belite**	Ca_2_SiO_4_	C_2_S	PDF 01-077-0409
**Alite**	Ca_3_SiO_5_	C_3_S	PDF 00-049-0442
**Ettringite**	Ca_6_Al_2_(SO_4_)_3_(OH)_12_(H_2_O)_26_	AFt	PDF 01-075-7554

**Table 2 ijms-24-07688-t002:** Semiquantitative analysis of the mineral phases and the degree of crystallinity of the prepared samples.

Sample	Mineral Phase (%)					Crystallinity Degree (%)
	CH	CC	C_2_S	C_3_S	AFt	
**CP**	17.3	15.6	42.8	24.2	-	71.8
**CPCa**	25.7	11.5	33.5	29.3	-	73.7
**CPNa**	16.7	13.8	33.1	36.5	-	65.2
**CPCaNa**	14.9	15.4	33.5	36.2	-	68.8
**CP2CaNa**	14.6	13.2	37.9	34.3	-	67.1
**CPCaNaAl**	12.3	13.0	31.3	32.4	10.9	67.4

**Table 3 ijms-24-07688-t003:** Pore size and distance between pores in studied samples.

Sample	Pore Identification Code	Pore Radius (μm)	Pore Area (μm^2^)	Distance between Pores (μm)
**CP**	C1	4.72	70.11	50.04
	C2	2.80	24.55	33.17
	C3	2.41	18.31	81.83
**CPCa**	C1	1.44	6.51	53.85
	C2	2.10	13.85	64.53
	C3	1.89	11.22	13.80
**CPNa**	C1	2.82	24.97	31.29
	C2	3.79	45.10	27.07
	C3	3.50	38.47	32.34
**CPCaNa**	C1	2.91	26.67	13.99
	C2	3.42	36.69	29.98
	C3	3.64	41.58	36.31
**CP2CaNa**	C1	2.62	21.55	16.63
	C2	2.37	17.64	17.96
	C3	2.55	20.42	14.41
**CPCaNaAl**	C1	3.37	87.38	16.86
	C2	2.27	46.82	22.23
	C3	3.22	25.19	10.48

**Table 4 ijms-24-07688-t004:** Elemental composition of samples’ map surfaces.

Element	Mass (%)					
	CP	CPCa	CPNa	CPCaNa	CP2CaNa	CPCaNaAl
**Oxygen**	52.65	50.31	50.25	51.84	50.67	49.12
**Calcium**	38.41	35.98	40.92	35.61	33.89	37.45
**Silicon**	7.52	7.91	5.92	6.53	7.23	6.09
**Aluminum**	1.29	1.33	1.05	0.94	1.33	1.34
**Sulphur**	1.19	1.46	1.92	1.91	1.34	2.41
**Sodium**	0.62	1.28	1.94	1.45	1.32	1.04
**Ca/Si**	5.10	4.54	6.91	5.45	4.68	6.14
**Ca/(Si + Al)**	4.35	3.89	5.87	4.76	3.95	5.04

**Table 5 ijms-24-07688-t005:** Mix proportion of the paste and abbreviation of the examined samples.

No.	Abbreviation	Concentration (wt%)			
		White Cement 52.5R	Ca(NO_3_)_2_·4H_2_O	Na_2_S_2_O_3_·5H_2_O	Al_2_(SO_4_)_3_·18H_2_O
1.	**CP**	100	-	-	-
2.	**CCPa**	97	3	-	-
3.	**CPNa**	97	-	3	-
4.	**CPCaNa**	97	1.5	1.5	-
5.	**CP2CaNa**	97	2	1	-
6.	**CPCaNaAl**	97	1	1	1

## Data Availability

Not applicable.

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
