# Peer review of "Structural Characterization of Several Cement-Based Materials Containing Chemical Additives with Potential Application in Additive Manufacturing"

_ijms, 2023, doi:10.3390/ijms24097688_

Round 1

Reviewer 1 Report

In general, as I concluded, the authors aimed to develope cement based materials for 3D printing?  Could it be tested for this purpose? 

I have some comments:

1. p.2.  line 61. : You write that there are "four main AM technologies:". Then you mention six of them.  Please, correct or explain it! Line 76.  in this page: the Ca(OH)2 is not pore product it is sludge like in wet state.  Line 83. it is written that SEM non-destructive. How did you make the SEM? Without coating?

P.3 line 110. please, define the unit of crystalline and amorphous state in order to understand what kind of percentage is the crystallinity. 

In Table 3 the radius or the diameter should be removed.They are not independent.

In line 201 RMN must be NMR.  In line 203 the slope is probably initial slope. In line there must be "chemically bonded water". In line 295 the SEM experiment should be described in larger extent (coated with gold or not).

As a summary the authors have proven that the combination of dry and wet studies give more significant results for application. It would be interesting to see that these results are correspond to the results of printing in 3D. Did you make any tests?

Reviewer 2 Report

The submitted paper (ijms-2321957: Structural characterization of several cement-based materials with potential application in additive manufacturing) is a kind of a very 'early stage' attempt to support some concretes' additive by a anemic (traditional basic science) framework (for 3D concrete printing). It is based on a weak skeleton combining some methods selected from a (certain, but) very limited list, that belong inside a traditional (basic science, but feeble) framework for this target (3D concrete printing). In reality, such a (useful) scientific 'product' (project), correlated with a "3D Concrete Printing", requires some additional (particular mechanical) methods in order to support a proposal (multi-physics) project about sustainable "materials for 3D concrete printing".
However, the submitted paper should include, at least, an additional method(s) should, also, support (established basic science) measurements  (vs. timeline of the composites' hydration) for some critical mechanical parameters (and other properties) of the composite materials and (multi-)layers, as building blocks/parts for some (captain) pilot 3D concrete actual-printing.
Also, the initial paragraph '3. Materials and Methods' should go/relocated before the paragraph "2. Results and discussion", e.g.  to be, as: '2. Materials and Methods'.

Reviewer 3 Report

The manuscript entitled Structural characterization of several cement-based materials with potential application in additive manufacturing offers the characterization of chemical additives to accelerate hydration of 3D printing mortar. Ca(NO3)2·4H2O, Al2(SO4)3·18H2O and Naâ‚‚Sâ‚‚O₃·5Hâ‚‚O were added for the acceleration purposes. The novelty of the manuscript seems promising; however, it is suggested to revise prior to accepting it. The comments include:

1. Title: the current title is not specified. The uses of several chemical additives should be mentioned in the title.

2. Keywords: cement-based material, chemical additive, hydration should be added

3. Introduction

               3.1 Line 31-38: The virgin resource extraction was mentioned. However, the work is not related to this aspect. The sentences should be removed.

               3.2 Line 65-74: the 3D printing cement accelerator aspects is needed to elaborate. Many chemical additives to increase hydration rate for 3D printing mortar and concrete were evaluated. The calcium aluminate cement has been widely used as 3D printing cement accelerator.  The relevant suggested references to review and cite include Lean manufacturing and thermal enhancement of single-layer wall with an additive manufacturing (AM) structure (ZKG International 2019); Mechanical properties of cement-based materials and effect of elevated temperature on three-dimensional (3-D) printed mortar specimens in inkjet 3-D printing (ACI Materials Journal 2019)

4. Line 100: 2. Results and discussion: correct this part

5. Line 277: Materials and Methods: correct this part

6. Line 278: The introduction and Table 5 discussed about Portland cement. But the White cement 52.5R was used. The verification and discussion were needed.

7. Line 283: The discussion why 3% of additive used should be elaborated

8. Line 284: Table 5. “Chemical composition” changes to “mix proportion of the paste”

9. Table 5: add Concentration (wt%)

10. Line 283: Elaborate regarding the ages of the tested sample and the stop hydration process

11. Line 299: What is the magnification level used?

12. Line 117: Figure 2 should be simple bar chart format

13. Experimental section: what is the method testing the pore size analysis?

14. Conclusion: The draw-up on the best additive among these three should be mentioned.

15. Reference: Provide suggested references

Reviewer 4 Report

For reviewing the manuscript ” Structural characterization of several cement-based materials with potential application in additive manufacturing” written by Alexandru Simedru, Anca Becze, Oana Cadar, Daniela Alexandra Scurtu, Dorina Simedru and Ioan Ardelean. The authors have worked about using the additions as: Ca(NO3)2·4H2O, Al2(SO4)3·18H2O, Naâ‚‚Sâ‚‚O₃·5Hâ‚‚O and combination between them in order to propose cement with fast hydration. However, when we read this paper from Title and the abstract we will wait that we will get results about 3D printing and some applications about that. When we read the introduction which is very very general without results and what is has been done as  results in literature. This introduction remined me when I give course to our students, how I introduce about 3D printing before starting the course. However, when we arrive to results and discussion (which is very poor in discussion and comparison to the literature) we feel lost and completely disconnected. The results concern only the structure and microstructure study of additions to the cement for fasting hydration. There is no 3D printing or application about 3D printing, where in introduction there is only 3D printing. There is big problem with this paper. In research article, all paper parts should be connected, not disconnected. This paper can be accepted for international conference but not as publication in journal. I am sorry, I reject it

Round 2

Reviewer 2 Report

This version is better. However, it is recommended to add the word 'studies', in the (new) title.

Reviewer 3 Report

The paper can be published in the current form.